# Exploring the Effect of Ethnicity on Chronic Orofacial Pain: A Comparative Study of Jewish and Arab Israeli Patients

**DOI:** 10.3390/healthcare11141984

**Published:** 2023-07-08

**Authors:** Robert Yanko, Yaara Badran, Shirley Leibovitz, Yair Sharav, Yuval Vered, Naama Keshet, Andra Rettman, Doron J. Aframian, Yaron Haviv

**Affiliations:** 1Department of Oral Medicine, Sedation & Imaging, Hadassah Medical Center, Faculty of Dental Medicine, Hebrew University of Jerusalem, Jerusalem 91120, Israel; 2In Partial Fulfillment of DMD Requirements, Hebrew University-Hadassah School of Dental Medicine, Jerusalem 91120, Israel; 3Department of Pediatric Dentistry, Barzilai Medical Center, Ashkelon 78306, Israel; 4Department of Community Dentistry, Hadassah Medical Center, Faculty of Dental Medicine, Hebrew University of Jerusalem, Jerusalem 91120, Israel

**Keywords:** chronic pain, orofacial pain, ethnicity, Arabs, Jews

## Abstract

The relationship between ethnicity and chronic pain has been studied worldwide. The population of Israel includes two main ethnic groups, 75% Jews and 21% Arabs. The purpose of this study was to compare orofacial chronic pain characteristics and treatment outcomes between Jewish and Arab Israeli citizens. Two hundred patients admitted to the Orofacial Pain Clinic at Hebrew University–Hadassah School of Dental Medicine between 2017 and 2022 were selected randomly for this historical cohort study. Our cohort included 159 (79.5%) Jews and 41 (20.5%) Arabs. Twenty-six pain-related variables were compared of which only two differed significantly between the two groups, awakening due to pain and mean muscle sensitivity; both indicators were higher in the Arab group (*p* < 0.05). No differences were found in any of the other variables such as diagnosis, pain severity, onset, and treatment outcome. This minimal difference may be explained by the equal accessibility to medical services for all citizens, and the diversity of our staff that includes Jew as well as Arab service providers. These factors minimize or even eliminate racial bias, language, and cultural barriers, and is reflected in the minor differences in orofacial pain characteristics found between the two main ethnic groups in Israel.

## 1. Introduction

Pain is an unpleasant sensory and emotional experience associated with actual or potential tissue injury, often described in terms of such damage [1]. Chronic pain is defined as persistent or recurrent pain lasting longer than 3 months, with physical, cognitive, and emotional consequences. Pain in the head and oral cavity space—orofacial pain (OFP) occurs in approximately 10% of the adult population [2]. Our clinic specializes in the comprehensive evaluation of chronic orofacial pain, which can be classified into three major groups: neuropathic [3], neurovascular (such as migraines), and musculoskeletal (such as TMD/temporomandibular pain) [4,5]. 

Pain experience can be influenced by ethnic factors like socioeconomic status, expectations language, spirituality, etc. [6,7,8,9].

There are numerous studies on the relationship between ethnicity and the nature of chronic pain [10,11,12,13]. For instance, African Americans reported stronger responses to painful stimuli, lower pain tolerance, and higher pain ratings following supra threshold stimuli than non-Hispanic Caucasians [12]. This may be due to social, cultural, psychological (e.g., coping, mood), and biological factors. In Australia, Italian men (minority) are more likely to report pain as frequent, severe, or chronic than non-Italian Australians [11]. Similar results are seen in a German study demonstrating a higher incidence of temporomandibular disorder (TMD) in the Chinese minority population than in the majority population [13]. Another study, for example, indicates contrasting treatment preferences among Jewish and Arab patients with painful inflammatory bowel disease. The study revealed that patients’ preferences were influenced by factors such as ethnicity, gender, and socio-economic disparities [14].

These studies emphasize the differences in pain characteristics, such as perception, frequency, severity, and spreading, between ethnic groups in terms of population majority vs. minority. Edwards et al. [10] demonstrated a higher sensitivity to pain in individuals from minority backgrounds compared to the rest of the population. The patients also responded differently to pain treatment, highlighting the need for tailored pain treatment approaches. Edwards et al. suggested that socioeconomic inequality between the majority and minority ethnic groups produced inadequate medical treatment in the latter, prolonging the sensation of pain. The authors also hypothesized that the pain treatments might also be influenced by clinician racial bias and stereotyping [10].

Israel is a multiethnic state known for its heterogenic population, with a majority of Jews (75%) and a minority of mainly Arabs (21%). Previous publications comparing the diagnosis of TMD in relation to Jews and Arabs in Israel revealed no differences regarding pain duration, intensity, and disability period, yet significant differences in chronic pain behavior and disability scores were recorded [15,16].

The aim of this study is to delve into the distinct features and treatment outcomes of chronic orofacial pain conditions among the Arab minority and Jewish majority populations in Israel. By conducting a comprehensive analysis, this research seeks to shed light on the potential variations in the nature and management of orofacial pain experienced by these two communities.

## 2. Methods and Materials

The study was approved by the Ethical Committee Hadassah Medical Center, request no. 329-21. The medical records of 200 OFP patients meeting our inclusion criteria attending the OFP Clinic at the Hebrew University–Hadassah School of Dental Medicine were reviewed. Patients were selected in a quasi-random manner based on their order of presentation to the clinic from February 2017 to June 2022 until the desired sample size of 200 patients was achieved. 

The number and proportion of Arabs and Jews in the cohort were matched to the population (approximately 75% Jews and 21% Arabs) and therefore the clinical records of 159 (79.5%) Jews and 41 (20.5%) Arabs were further studied. The data were anonymized, with all members of the cohort given a reference number.

Inclusion criteria: over 18 years of age; diagnosis of chronic, non-dental-related OFP, for at least 3 months; definite ethnicity; at least one recall visit.

The exclusion criteria for this study included patients with orofacial pain lasting less than 3 months, pain due to dental causes, patients with unclear origin or ethnicity, patients with no recall visits, patients with a history of drug abuse, and pregnant or lactating patients. Additionally, patients whose pain was found to have a systemic pathologic cause based on history, examination, or imaging were also excluded. All diagnoses were confirmed in the clinic and later re-examined by the senior author (YH) after data tabulation and summary.

### 2.1. Data Collection

Primary and resultant data were taken from the standard intake form used in our clinic [17,18,19]. Demographic data included ethnicity (Jewish or Arab), sex, and age.

Pain characteristics included onset (months); laterality, i.e., whether the pain is unilateral or bilateral; whether the pain is constant, comes in attacks or a combination; pain quality—burning, electrical, pressure, throbbing, and stabbing sharp; pain intensity rated using an 11-point verbal pain scale (VPS), where 0 is no pain and 10 is the worst pain imaginable [20]. Health-related quality of life (QoL) over the last month on a 0–10 numeric scale was also recorded [17]. Patients were asked specifically whether their pain wakes them from sleep (not according to sleep test). Pain that began following a traumatic event was defined as “post-traumatic” and divided into macro-trauma (road traffic accidents and altercations) and micro-trauma (dental surgery: invasive or prolonged interventions). General pain syndromes (e.g., fibromyalgia, chronic headaches), accompanied systemic (e.g., nausea) or autonomic signs (tearing) were recorded, as were prescribed pain treatments. The medications were categorized as (1) Tricyclic’s/SNRIs; (2) anticonvulsants; and (3) “others”—any medication not in the first two categories; (4) “combination” for when several medications were used concomitantly. Conservative treatments such as night guard or physiotherapy were recorded as “no drugs”.

Clinical examination included masticatory apparatus palpation, as previously described [21,22,23]. Briefly, total muscle sensitivity was measured by applying pressure (1–2 kilos) on two muscles related to chronic pain in the orofacial area; masseter and temporalis on both sides. The patients scored their sensitivity on a scale of 0–3 (3 being the highest sensitivity level). The clinician documented the degree of sensitivity for each muscle (left/right masseter, left/right temporalis). The sum of the sensitivity scores termed “Total muscle score” was between 0 and 12.

### 2.2. Patient Groups Based on Orofacial Pain Diagnosis

Musculoskeletal Group—Temporomandibular Disorders (TMD) based on the diagnostic criteria for temporomandibular disorders (DC/TMD), often associated with pain in the pre-auricular region and/or masticatory muscles, TMJ and mandibular movement dysfunction [4].Neurovascular Group—including migraine (as well as facial migraine and NVOP [3] and tension type headache (TTH) according to ICHD-3 criteria [24].Trigeminal Neuralgia (TN) [18]—neurological pain characterized by recurrent unilateral brief electric shock-like pain, abrupt in onset and termination, along the distribution of the trigeminal nerve and triggered by innocuous stimuli [24].Neuropathic Group—Post-traumatic Trigeminal Neuropathy (PTN) [24], unilateral facial or oral pain following trauma to the trigeminal nerve. [3] Burning Mouth Syndrome (BMS), a chronic condition characterized by a moderate to severe sensation of burning from the oral mucosa, especially form the dorsum of the tongue with no clinical signs, and PIFP—Persistent Idiopathic Facial Pain with neuropathic elements [3].

### 2.3. Statistical Analysis

SPSS software version 22.0 was used for the analysis. Continuous variables were described as means and standard deviations, while categorical variables were described as absolute numbers and percentages. Two-tailed statistical significance (α) level was defined as *p* < 0.05.

Descriptive analysis was used for qualitative variables and presented in absolute numbers and percentages (Table 1). Means and standard deviations were used for quantitative variables (Table 2). Statistical significance was set at *p* < 0.05. The following statistical tests were used for nominal variables, Chi-square significance test was performed, and for ordinal variables, non-parametric Mann–Whitney and Wilcoxon rank tests were performed. For normally distributed continuous variables, t-test and Pearson correlation were performed, and for non-normally distributed variables Spearman correlation and non-parametric Mann–Whitney and Wilcoxon rank tests were used.

## 3. Results

This study included 200 patients from the two largest ethnic groups in Israel, 159 (79.5%) Jews with a mean age of 48 ± 17.20 years old and 41 (20.5%) Arabs with a mean age of 44.3 ± 15.61. The study included 66 (33.3%) males and 164 (66.6%) females. The Jewish group had 53 (33.3%) males and 106 (66.6%) females, and the Arab group had (31.7%) males and 28 (68.3%) females.

The distribution of specific pain diagnoses in each ethnic group is provided in the Appendix A.

The only variable that was significantly different between the ethnic groups was total muscle score. The score of (6.0897 ± 9.32023) in the Arab group was significantly higher than that of the Jewish group (2.8547 ± 3.3938).

When comparing primary VPS and final VPS means, the Jewish primary VPS and final VPS means (8.1644 ± 8.73 and 4.7368 ± 4.88, respectively; *p* > 0.05) were both higher than Arab primary VPS and final VPS means (7.5 ± 1.76 and 4.1 ± 2.50, respectively; *p* > 0.05). The differences between the groups were not statistically significant; the changes in VPS within each group showed that there were significant improvements from the treatments given for both groups. It is important to note there are other indices for success such as improved function and restored social functioning that were not assessed in this study.

No significant differences were found between the two ethnic groups for any of the other variables examined (*p* > 0.05) (see Table 1).

## 4. Discussion

During the last two decades, the number of academic publications focusing on the racial and ethnic differences in the experience of chronic has grown [17]. Some studies reported higher pain prevalence in minority groups, while others found no differences in chronic pain prevalence based on race/ethnicity [25,26,27].

In light of the inconsistency, we compared the prevalence of a wide range of OFP entities and treatment outcomes between the two main ethnic groups that are citizens of Israel, namely the Jewish population and the largest minority group, the Arab population.

Although the differences did not reach statistical significance, neuropathic pain and musculoskeletal pain were more common in the Arab group, whereas the Jewish group had higher incidences of neurovascular pain and TN (12% of Jews versus 2% of Arabs). It is possible that the small sample size caused the statistical insignificance.

Pain types, especially with hereditary components, are more common in some populations; for example, migraine prevalence is lower in Asians compared to other ethnic groups [28]. While there is contradictory information regarding pain prevalence in ethnic minorities, most studies suggest that the severity and impact of pain on daily life is greater among minorities with chronic pain [25,29,30]. In contrast to most publications, we did not find a higher pain intensity in the Arab group; on the contrary, the Jewish group reported insignificantly higher pain intensity levels. The study by Riley et al. [31] supports our findings that ethnicity has little impact on pain intensity. However, the authors noted that the differences are expressed at a later stage of pain processing, including the emotional and behavioral responses associated with chronic pain. Similarly, a previous Israeli study on TMD reported no differences in pain intensity between Arabs and Jews and significant differences in other parameters like depression, anxiety and somatization [16]. This highlights the need for further investigation of those parameters.

There is a growing body of evidence that ethnicity influences the response to experimentally induced pain [31]. Studies found that sensitivity to noxious stimuli is enhanced in African Americans and Asians compared to Caucasians [29,32,33,34]. This may be reflected in the higher total muscle score in the Arab group. More Arabs reported that their pain awakens them from sleep. The bidirectional relationship between chronic pain and psychosocial stress is well established. Individuals who experience adverse life events have a greater risk of developing chronic pain conditions. Moreover, living with chronic pain is associated with increased psychosocial stress. Minority ethnic groups are more susceptible to sleep disturbances and sleep awakening probably due to higher levels of environmental stressors such as economic hardships, discrimination, and limited access to healthcare [35]. In this study, we did not focus on the precise mode of treatment for each diagnosis. This comparative study aimed to evaluate differences between two populations across various aspects.

This study has several limitations including its retrospective nature and lack of data on socio-economic variables such as education level, income, family status, and employment status. There are several factors that may influence pain characteristics, diagnosis, and treatment in addition to specific ethnicity, such as prior treatments the patients may have undergone, employment status, profession, level of education, marital status, financial status, religious beliefs, etc. However, for the purpose of this study, we have chosen to focus solely on ethnicity-specific factors.

Differences in pain characteristics between majority and minority groups may be attributed to expectations for pain relief [9], social inequality and racial bias. It was suggested that this inequality leads to inadequate medical care and prolonging the sensation of pain [35]. Edwards et al. [10] hypothesized that clinician bias and racial stereotypes may influence the treatments provided.

In contrast to these findings, we only discovered minor differences in pain characteristics between the Jewish and Arab populations in the current study. These may be attributed to several factors:The National Health Insurance program in Israel aims to provide equal access to medical services for all citizens, but there may be variations or limitations in practice.A recently published report noted that the health of the Arab population in Israel is improving at the same rate as the improvements in the health status of the Jewish population, and that the health challenges of populations in the social and geographic periphery of the state are the same for Arabs and Jews [36].As an academic institution, the Faculty of Dental Medicine of the Hebrew University and Hadassah Medical Center complies with the vision, mission and values of “perceiving diversity as a condition for academic excellence and for realizing the human potential of the Israeli society” and “treating all patients without discrimination or bias, and maintaining fairness and respect for patients and staff members (equality)”. The dental faculty includes undergraduate and postgraduate students and residents of all ethnicities and diversity and, as such, minimal racial bias and stereotyping, as well as language and cultural barriers exist between the patients and the caregivers.

In conclusion, orofacial pain is a common condition that affects a significant portion of the population of all ages, ethnicities and genders. The demographics of OFP may have an impact on the quality of life, daily functioning, and overall well-being of the individual patient. No significant differences were found between Arab and Jewish population in Israel regarding OFP diagnosis and characteristics. However, pain in general is a complex and multifaceted phenomenon that can be influenced by a wide range of factors, including cultural and social factors, as well as individual differences in pain sensitivity and coping strategies. The study is exploratory in nature, which means that we cannot prove the hypotheses based on the results. While our research provides valuable insights into the potential role of ethnicity in orofacial pain, further research using more specific and systematic methods may be needed to confirm our findings and establish causal relationships.

## Figures and Tables

**Table 1 healthcare-11-01984-t001:** Pain and patient characteristics.

Variable	*N*	Total Mean ± SD	*N*	Jews Mean ± SD	*N*	Arabs Mean ± SD	*p*-Value Independent Sample *T*-Test
Age	200	47.22 ± 17	159	48 ± 17.20	41	44.30 ± 15.61	(t(198)) = 0.096, *p* > 0.05)
Onset (Months)	199	86.65 ± 203.07	158	81.73 ± 19.02	41	105.60 ± 236.44	(t(197)) = −0.671, *p* > 0.05)
Sleep quality	124	8.39 ± 11.92	98	9.24 ± 13.20	26	5.21 ± 3.27	(t(122)) = 1.538, *p* > 0.05)
QoL	91	5.84 ± 6.07	75	5.97 ± 6.53	16	5.19 ± 3.25	(t(89)) = 0.467, *p* > 0.05)
Unassisted opening (Millimeters)	148	43.49 ± 8.82	117	44.03 ± 9.27	31	41.45 ± 6.61	(t(146)) = 1.454, *p* > 0.05)
Primary–VPS	179	8.03 ± 7.92	146	8.16 ± 8.73	33	7.50 ± 1.76	(t(177)) = 0.464, *p* > 0.05)
Final–VPS	66	4.66 ± 4.62	57	4.74 ± 4.88	9	4.10 ± 2.50	(t(64)) = 0.636, *p* > 0.05)
Time from first to last recorded meeting (months)	171	24.11 ± 40.77	141	25.13 ± 44.1	30	19.28 ± 19.08	(t(169)) = 0.156, *p* > 0.05)
Total muscle score	187	3.53 ± 5.34	148	2.85 ± 3.39	39	6.09 ± 9.32	(t(40.689)) = −2.13, *p* < 0.05)

QoL = Health-related Quality of Life. Primary VPS scored on a pain scale (0–10) and reported by the patient during their initial visit. Final VPS scored on a pain scale (0–10) and reported by the patient on their last recall appointment. Time from first to last recorded meeting = time (in months) from the first meeting to the last meeting where the patient reported a change in their pain perception. Total muscle score = the sum muscle sensitivity scores. Sleep Quality—measured on a numerical 11-point (0–10) scale for a specific question related to sleep quality [19,25]. Independent sample *T*-test: Absolute numbers and percentages can be lower than patient cohort if some answers were missing or ambiguous.

**Table 2 healthcare-11-01984-t002:** Diagnosis and pain characteristics.

Variable	Total *N* (%)	Jews *N* (%)	Arabs *N* (%)	*p*-Value (Chi-Square Test)
**Sex**				(χ2 (1) = 0.039, *p* > 0.05)
Males	66 (33%)	53 (33.3%)	13 (31.7%)	
Females	134 (67%)	106 (67.6%)	28 (68.3%)	
**Primary Physical Trauma**				(χ2 (2) = 2.012, *p* > 0.05)
None	105 (53%)	81 (51.6%)	24 (58.5%)	
Micro	55 (27.8%)	49 (31.2%)	6 (14.6%)	
Macro	37 (18.7%)	26 (16.6%)	11 (26.8%)	
** Laterality **				(χ2 (2) = 5.781, *p* > 0.05)
Unilateral	121 (61.1%)	96 (60.8%)	25 (62.5%)	
Bilateral	77 (8.9%)	62 (39.2%)	15 (37.5%)	
**Awakening from sleep**				** (χ2 (1) = 5.507, *p* < 0.05) **
Non-awakening	110 (57.6%)	94 (61.8%)	16 (41%)	
Awakening	81 (42.4%)	58 (38.2%)	23 (59%)	
Pain-Related Disorder	59 (29.5%)	43 (27%)	16 (39%)	(χ2 (1) = 2.249, *p* > 0.05)
**Pain Quality**				
Pressure	114 (59.4%)	92 (62.9)	22 (58.6)	(χ2 (2) = 5.025, *p* > 0.05)
Throbbing	39 (20.3%)	34 (21.7%)	45 (14.3%)	(χ2 (1) = 4.504, *p* > 0.05)
Burning	45 (23.4%)	41 (26.1%)	4 (11.4%)	(χ2 (4) = 4.02, *p* > 0.05)
Stabbing	47 (24.5%)	37 (28.6%)	10 (23.6%/0	(χ2 (2) = 5.092, *p* > 0.05)
Electrical	17 (8.8%)	14 (8.9%)	3 (8.3%)	(χ2 (1) = 5.056, *p* > 0.05)
**Systemic or Autonomic signs**				(χ2 (2) = 5.245, *p* > 0.05)
None	129 (64.2%)	110 (69.2%)	19 (46.3%)	
Nausea	4 (2%)	4 (2.5%)	0	
Vomiting	1 (5%)	0	1 (2.4%)	
Photophobia	5 (2.5%)	4 (2.5%)	1 (2.4%)	
Phonophobia	6 (3%)	6 (3.8%)	0	
Dizziness	11 (5.5%)	4 (2.5%)	7 (17.1%)	
Combination	411 (20.5%)	30 (18.9%)	11 (26.8%)	
**DIAGNOSIS**				(χ2 (2) = 5.52, *p* > 0.05)
Musculoskeletal	68 (34%)	51 (32.1%)	17 (41.5%)	
Neurovascular	51 (25.5%)	43 (27%)	8 (19.5%)	
Neuropathic	60 (30%)	45 (28.3%)	15 (36.6%)	
Trigeminal Neuralgia	20 (10%)	19 (11.9%)	1 (2.4%)	
Others	1 (0.5%)	1 (0.6%)	0	
**Treatment**				(χ2 (3) = 5.021, *p* > 0.05)
Tricyclic’s\SNRIs	51 (27.6%)	36 (24.3%)	15 (40.5%)	
Anti-Convulsions	37 (20%)	29 (19.6%)	8 (21.6%)	
Other	56 (30.3%)	48 (32.4%)	8 (21.6%)	
No Medications	21 (11.4%)	15 (10.1%)	6 (16.2%)	
Combination	20 (10.8%)	20 (13.5%)	0	

Pain-related disorder—history of pain-related disorder (including chronic headaches or fibromyalgia). Physical trauma causing or affecting the chronic pain condition was classified as (i) none meaning no trauma; (ii) micro such as dental procedure; (iii) macro such as a car accident. Laterality describes pain location: unilateral or bilateral. Awakening from sleep describes whether the pain awakens the patient from sleep. Pressure, throbbing, stabbing, electrical or burning description and perception of the pain, reported as yes/no for each term. Systemic signs phenomena occurring with the pain. Diagnosis and medications: see methods section. Chi-square test: Absolute numbers and percentages can be lower than patient cohort if some answers were missing or ambiguous.

## Data Availability

Data sharing not applicable due to privacy or ethical restrictions.

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
