# Peer review of "Exploring the Effect of Ethnicity on Chronic Orofacial Pain: A Comparative Study of Jewish and Arab Israeli Patients"

_healthcare, 2023, doi:10.3390/healthcare11141984_

Round 1
Reviewer 1 Report (Previous Reviewer 1)
Paper is acceptable.
English is acceptabel.
Author Response
Thank you. for the very positive response
Reviewer 2 Report (Previous Reviewer 3)
1. Introduction: there is an extra space at the beginning of line 43. There should be an indent at the beginning of line 62. Also, I wondered if there are more up-to-date references for you to include in your introduction.
2. There is an inconsistency in the number of participants you included in this study.
In lines 72-73, "the clinical records of 161 (80.5%) Jews and 39 (19.5%) Arabs were further studied." However, in lines 138-139, "159 (79.5%) Jews with mean age of 48±17.20 years old and 41(20.5%) Arabs with mean age of 44.3± 15.61." I wondered which one is correct.
4. Did you collect other sociodemographic factors such as educational levels, income levels, etc.?
3. Table 1 and results: I would suggest just keeping two decimal point results in your table.
4. You may need more sentences in your result section. I am not sure why the "Total N" in Table 1 for each variable is different. What does the "success rate" mean? Is the unit of "time from first to last recorded meeting" month or day?
5. lines 241-250. The font and size are inconsistent with other paragraphs.
Round 2
Reviewer 2 Report (Previous Reviewer 3)
The manuscript context has been improved. I believe there is room for further improvement, but I understand that it would be difficult to collect data retrospectively given the nature of the interview study.
This manuscript is a resubmission of an earlier submission. The following is a list of the peer review reports and author responses from that submission.
Round 1
Reviewer 1 Report
As noted, the sample size could be improved and any conclusions should be interpreted with caution. Adding socio-economic variables would certainly strengthen the paper. Particularly with the retrospective nature of this study, Increasing the sample size of the Arab group would be warranted.
However, as written, the paper can stand own its own merit.
Reviewer 2 Report
Dear Editor,
Thank you for forwarding this interesting article to me for review.
The article is based on a sample from an Israeli population, with the objective of assessing the influence of ethnicity on chronic orofacial pain. It is a retrospective clinical study from which a series of considerations have come to my attention, which I will now comment on for the authors, in case they would like to take them into account in order to improve their article.
Lines 19-20 and 68-70, and in the Title. Study design. The authors define their study as a cross-sectional observational study, with randomly patient selection. I have some doubts about this definition that I would like the authors to clarify.
Strictly speaking, we are dealing with a retrospective study carried out on a review of medical records where two cohorts are compared. The authors use the term "patients randomly selected". This term is misleading since what is actually being reviewed are consecutive patient records in reverse chronological order. It is also not exactly a cross-sectional study, as the data are not collected in a specific time frame. The data are collected over a broad period of years, in a descriptive manner. It would be good if the authors could clarify these confusing aspects for the reader.
The study period covers the years 2022 to 2017. Therefore, it includes periods of confinement due to the coronavirus pandemic. I wonder if health restrictions during the months of confinement could have affected data collection and patient management.
Lines 76-77. The exclusion criteria are defined in negative, i.e. in contrast to the inclusion criteria. More extensive exclusion criteria are lacking.
Lines 85-86. Quality of life data were collected, apparently using a simple numerical scale from 0 to 10. Why were not used internationally validated quality of life questionnaires with qualitative scales?
Line 79 and following. Primary patient data were taken at the time of the first visit. It is striking that the previous treatments that the patients had undergone are not known. What were the previous treatments that the patients had undergone for their chronic orofacial pain? In this sense, the previous treatments they were undergoing at the time of diagnosis could have biased the data from the first visit.
On the other hand, a weakness of this study is that the protocol of treatment for chronic orofacial pain carried out in the institution is not explained. For example, it is striking that most of the patients had been diagnosed with musculoskeletal pain (34%). The main management modalities currently available for the management of musculoskeletal pain are patient education and control of predisposing factors, analgesics, pharmacological therapies such as myorelaxants and antidepressants, occlusion splints, physiotherapy treatments, dry needling, ultrasound, transcutaneous electrical nerve stimulation, psychotherapy, or muscular injections and wet needling of different substances such as anaesthetics or botulinum toxin. However, reference is only made to the use of night splints and physiotherapy. Is this so? If so, I feel that patients with musculoskeletal pain were poorly treated. There are missing lines of treatment and procedures that were not being performed. To determine the success rate of treatment in both cohorts, it would be necessary to establish a defined treatment protocol that all patients will perform systematically.
Line 86 and table 1. How is sleep quality measured? It seems to be a scale from 0 to 10. Could the authors please specify how this measurement is carried out?
Table 1 shows the success rate and gives a numerical value. The measurement was not explained, nor was it explained when or at what checkpoint in posttreatment this measurement was made. The success criteria are also not detailed. It seems that the success criteria are limited only to pain as measured by the VAS scale at the last visit. However, the success criteria in orofacial musculoskeletal temporomandibular pain include other factors such as mandibular function (range of opening and closing of the mouth at rest or in movement), joint locking, etc. Were these parameters of therapeutic success taken into account?
In table 2, in the data on diagnosis of chronic orofacial pain, in the Arab group 41.5 % of musculoskeletal pain is reflected. However, in the treatment data for the Arab group, only 16% did other non-medication treatments. This is confusing, as chronic orofacial musculoskeletal pain responds poorly to drug treatment, and other procedures including minimally invasive procedures should be used.
Table 2 also refers to sleep awakening, and is divided into non-awakening and awakening, which is rather subjective.
Figure 2 is redundant to table 2. Furthermore, it is confusing to the reader because figure 2 does not give decimal numbers, and this means that the data do not exactly match the data in table 2 which have decimal numbers. I would advise deleting figure 2 and referring the reader to table 2. Incidentally, in the copy of the paper that has reached me, I cannot find the reference to figure 1.
Line 218 and following. The authors discuss the limitations of this study. This study has the significant limitation that it does not provide important sample data that may influence chronic orofacial pain and bias the results. This study has the important limitation that it does not provide data that could influence chronic orofacial pain and bias the results. For example, we do not know what previous treatments the patients had undergone. Nor do we know data on employment status, profession, level of education, marital status, financial status, religious beliefs, etc. Numerous data are missing, which impact chronic orofacial pain to a greater extent than even ethnicity. We also do not know how the patients completed the treatment, possible adverse effects, etc.
Line 227 and following. The authors comment on some factors that may have influenced the study, namely the National Health System and improvements in the health status of the Israeli population. All these arguments are speculative because they are not based on data collected from medical records. In any case, these are arguments that may be valid internally and locally where the research is conducted, but are of doubtful external validity at the international level.
The Discussion section lacks a broader discussion of the effects of therapeutic measures on chronic orofacial pain. It is not concluded how the treatments performed impacted the results, what effect they had, and what success rate they had on long-term chronic orofacial pain.
Finally, I cannot find if the study received specific funding.
Reviewer 3 Report
This study examined the role of ethnicity (Jewish and Arabs) in orofacial pain at an OFP clinic in Isreal. I have some suggestions and concerns about this study.
A. Consistency
1. In the Methods and Materials section, the authors abbreviated the verbal pain scale as VPS, but they presented this variable as VAS in the tables and write-up. Also, from lines 122 to 124, the authors stated that Table 1 would show the absolute numbers and percentages. Yet, the Table presented is the other way around.
2. The study title indicated that the "treatment success" will be presented. However, I am not aware of where the treatment success was described except that the "success rate" was presented in Table 1. However, I am not clear about what the success rate means.
B. Methods:
1. I wondered how the authors calculated their sample size (i.e., 200 patients). Was it a convenient sampling? Perhaps, the authors could describe their effect size.
2. In Table 1, the SD of "onset" variable for the Total and Arabs columns is wide. Maybe showing Median would be more appropriate.
3. This study is exploratory without showing a model result. Therefore, it would not be sufficient to see and conclude the role of ethnicity in OFP.